# Patent Foramen Ovale in Fetal Life, Infancy and Childhood

**DOI:** 10.3390/medsci8030025

**Published:** 2020-07-01

**Authors:** Bibhuti B. Das

**Affiliations:** Department of Pediatric Cardiology, Baylor College of Medicine, Texas Children’s Hospital Specialty Care Austin, Austin, TX 78759, USA; bbdas@texaschildrens.org; Tel.: +17-37-220-8328; Fax: +17-37-220-8180

**Keywords:** foramen ovale, patent foramen ovale (PFO)

## Abstract

A patent foramen ovale (PFO) is a common, incidental echocardiographic finding in otherwise healthy and asymptomatic infants and children. However, a variety of clinical conditions have been ascribed to the presence of a PFO in childhood, such as cryptogenic stroke, platypnea-orthodeoxia syndrome, decompression sickness and migraine, although the data on these are controversial and sometimes contradictory. This review discusses embryology and correlation with post-natal anatomy, anatomical variations of the atrial septum, diagnostic modalities in special circumstances of PFO associated clinical syndromes, and the role of PFO in congenital heart disease, pulmonary hypertension, dilated cardiomyopathy and heart failure in children who require an extracorporeal membrane oxygenator or ventricular assist device as life support.

## 1. Introduction

A patent foramen ovale (PFO) is a commonly discovered potential opening between the right atrium (RA) and left atrium (LA) on routine echocardiographic surveillance in an otherwise asymptomatic infant or child (Figure 1A,B). Screening for a PFO in an asymptomatic patient or follow-up for an isolated PFO is not justified [1]. There are limited data on the underlying causes of a PFO, although there is some good evidence that genetic factors play a role [2]. The prevalence of PFO in the general population is 25% and the median diameter of a PFO is 5 mm (mean ± standard deviation, 4.9 ± 2.6 mm); very rarely, a PFO can enlarge on follow-up [3,4]. There has been much discourse about the differences between a benign PFO and a pathological secundum atrial septal defect (ASD). The criteria to differentiate these two are the size of the hole, the presence or absence of a flap of septal tissue and the size of the interatrial shunt [5]. A systematic search of “foramen ovale (FO)”, “FO in fetal life”, “PFO”, “high risk-PFO in children”, “diagnosis of PFO”, “PFO and congenital heart defect (CHD)”, “PFO and migraine”, “stroke/transient ischemic attack (TIA) in children”, “platypnea-orthodeoxia”, “residual PFO after CHD repair”, “PFO and decompression illness”, “PFO and hypoxemia”, “PFO and thromboembolism” and “treatment of PFO in children” was performed through the PubMed and Cochrane databases to obtain information relevant to children, which is included in this review. Randomized control trials (RCT) comparing the treatment of PFO using either devices or medical therapy in children with many PFO-associated syndromes are not available. PFO-associated clinical syndromes are rare, most studies are observational, and it is difficult to inform a relative risk, absolute risk, odds ratio or a *p*-value of the disease associations with PFO based on which a definite clinical recommendation can be made. Existing guidelines are mainly for adults with cryptogenic strokes in the setting of PFO, and many trials are still on-going in adults. It is not ideal to extrapolate adult studies to children as there is a significant difference in the physiology contributing to thrombus formation and paradoxical embolism in adults compared to children. There is a need to conduct RCT of the role of PFO in clinical syndromes in children.

The importance of a PFO in some critical congenital heart defects (CHD), especially in neonates, is well recognized. However, there is controversy regarding the clinical significance of a PFO with vascular events [6,7,8], specifically when accompanied by anatomic anomalies [9] such as atrial septal aneurysms, Chiari networks and prominent Eustachian valves. A PFO associated with CHD is common and the physiology is very different from an isolated PFO [10]. The presence of a PFO has also been implicated in the pathogenesis of medical conditions such as platypnea-orthodeoxia syndrome [11], decompression sickness [3] or migraine [12] in children. Currently, there are no accepted guidelines for the management of PFO in the pediatric age group. This review describes the embryology, anatomy, variations of atrial septum associated with PFO, the relevance of PFO in fetal life, diagnostic modalities in special circumstances when a PFO is associated with CHD or other clinical scenarios during infancy and childhood and outlines an overall approach to the management of PFO-associated syndromes.

## 2. PFO in Fetal Life

In the embryonic stage, between 4 and 5 weeks of gestation, the septum primum begins to grow from the posterosuperior region of the primitive common atrium and grows in the midline towards the endocardial cushions at the crux of the heart. Smaller fenestrations develop in the mid-portion of the septum primum, even as the most inferior part merges with the developing endocardial cushions. The septum secundum also arises from the posterosuperior region of the common atrial wall, just to the right of the septum primum, and grows towards the endocardial cushions. Its inferior edge is concave and forms the crista dividens (limbus of the fossa ovalis). A second orifice forms as a flap valve mechanism in the upper portion of the septum primum (Figure 2A), which remains patent during fetal life as an FO (Figure 2B).

During fetal life, the Eustachian valve on the leftward ridge of the inferior vena cava facilitates the shunting of oxygenated blood from the umbilical vein to the left atrium (LA) via FO. This allows more oxygenated blood to return to the left ventricle (LV). This physiological right to left shunt in fetal life is large, unobstructed and essential [13]. At birth, the instant increase in pulmonary blood flow due to the onset of spontaneous ventilation is associated with increased venous return to LA. The distension of the LA with higher volume, along with the decreased vena caval flow into RA after cord clamping, results in the movement of the primum septum rightwards and allows the apposition and subsequent fusion of the primum with the secundum membranes. A deficiency in the membranous portion of the primum and the limbus results in a PFO (Figure 2C). 

The FO in fetal life can be reliably demonstrated by echocardiography using ultrasound modalities such as 2D, color and spectral Doppler during the second half of pregnancy [14]. The fetal FO is however difficult to demonstrate circumferentially despite advances in ultrasound imaging; hence, the transverse diameter is commonly used [14]. Fetal premature closure or restriction of the FO has been reported at any gestational age with variable effects on fetal hemodynamics and adverse fetal and postnatal outcomes. The spectrum of clinical presentation is related to the gestational age when the restriction first occurs and the severity of restriction. Significant flow restriction results in an additional amount of blood being re-directed to the right ventricle (RV) as it bypasses the LA, causing right heart failure manifesting as pleural/pericardial effusions, ascites, necrotizing enterocolitis and non-immune hydrops fetalis. There is also an association between the premature closure of the FO with neonatal mortality and complications such as prematurity, maternal preeclampsia and placental abruption [15].

Premature FO closure has been postulated as a cause of hypoplastic left heart syndrome (HLHS) [16] and neonatal pulmonary hypertension [17]. The diagnostic criteria for restrictive fetal FO include an FO diameter of <2 mm, a Doppler velocity >1.2 m/s, RA and RV dilatation, tricuspid valve regurgitation, pericardial effusion and/or hydrops fetalis [14]. More advanced quantification using color and spectral Doppler in the pulmonary artery and pulmonary veins, using the pulsatility index, is useful to monitor FOs [18]. Restriction of a fetal FO can however be difficult to ascertain in the presence of an atrial septal aneurysm because the fossa ovalis may appear widely patent. Close surveillance using fetal echocardiography could help determine progression, prognosticate outcomes and hence guide the management of the fetus and infant. Fetal cardiac intervention is an evolving management tool that has been used in HLHS with an intact or restrictive FO to alleviate or reduce the hemodynamic consequences [19]. The risk assessment of the fetus and the mother influences the optimal timing and method of delivery of the infant [20].

## 3. PFO in Neonates and Infants

The hemodynamic changes after birth lead to a smooth transition of fetal circulation and primarily depend upon the drop in pulmonary vascular resistance (PVR) and an increase in systemic vascular resistance (SVR). The presence of PFO plays an important role in understanding the normal cardiovascular transition physiology and its adverse adaptation. The shunt across the PFO is either left-to-right or bidirectional but never purely right-to-left. A pure right-to-left shunt across the PFO should raise suspicion of underlying CHDs such as total anomalous pulmonary venous drainage or tricuspid atresia. Assessment of the flow across the PFO, especially when it is bidirectional, may indicate persistent pulmonary hypertension of the newborn, and it contributes to monitoring in premature infants and infants with perinatal asphyxia.

In neonates with right heart obstructive lesions, such as tricuspid or pulmonary atresia, and left heart obstructive lesions, such as HLHS, the continued patency of the FO is critical so that an obligatory right-to-left or left-to-right shunt, respectively, across the atrial septum, can take place. In neonates with transposition of the great arteries (TGA), the circulation is parallel (instead of normal in-series circulation) and mixing across the circulations is essential for survival; this is usually provided by the ASD/PFO. With TGA, it is important to assess the atrial level shunt and a PFO ≤2mm warrants an urgent balloon atrial septostomy to improve survival [21]. The atrial septum in infants with HLHS can be very thick and, if PFO is restrictive, these newborns become critically ill [22]. In this case, instead of a balloon atrial septostomy, a blade septostomy or the creation of a separate defect in the atrial septum by Brockenbrough (needle atrial transeptal) puncture or radiofrequency wire perforation is needed.

## 4. PFO in Childhood

A PFO is frequently detected during childhood by transthoracic echocardiography (TTE) for the evaluation of a murmur or other medical condition. In the majority of children, TTE with or without contrast saline is sufficient to diagnose a PFO because of the excellent acoustic windows [23]. The most ideal window to visualize the PFO is the subcostal acoustic window. This is because the septum is relatively perpendicular to the transducer and adequately echo-reflective from that position. This reduces the likelihood of a false dropout and a misdiagnosis of a PFO. The confirmation of the PFO is required by documenting evidence of the transseptal flow by color Doppler. Both the subcostal long-axis and short-axis views can be utilized for a confirmatory diagnosis. The apical views often have false dropouts of the acoustic signals due to the parallel orientation of the transducer beam and the atrial septum position. This is not the most ideal location for the evaluation of a PFO. However, the apical-4 chamber view is helpful during the saline contrast “bubble” study in looking for the appearance of acoustic reflective signals (“bubbles”) in the LA and LV (representing right-to-left flow). There is also a need for the standardization of PFO identification and quantification for a saline contrast bubble study, which at present does not exist. In the French PFO- Atrial Septal Aneurysm (ASA) study, a PFO is identified if at least three bubbles appear in the LA. The degree of shunting is defined to be small if three to nine bubbles appear, it is moderate if 10–30 bubbles appear and large if >30 bubbles appear in the LA [24]. According to PFO in cryptogenic stroke study (PICSS), a PFO is considered to be present if more than one bubble appears in the LA and if >10 appear in the LA, it is considered as a large PFO [25]. Recently, it was shown that for a given PFO, the amount of right-to-left contrast shunting is a matter of expiratory pressure during the Valsalva maneuver [26].

Trans-esophageal echocardiogram (TEE) is traditionally considered the gold standard for detection of a PFO in the setting of stroke in the adult patients [27,28]. The TEE is necessary for children in order to delineate the anatomy of the interatrial septum during a surgical or interventional cardiac catheterization procedure. In some surgeries, such as tetralogy of Fallot (TOF) repair, a small PFO may be left behind or a punch hole made in the interatrial septum to allow right-to-left shunting and maintain cardiac output in the context of a hypertrophied RV with poor relaxation or potential increased afterload from smallish pulmonary arteries. In these situations, a right-to-left shunt or a bidirectional atrial shunt can be seen on the color Doppler. In other situations, such as a Shone complex with small left-sided structures, there is left-to-right atrial shunting at higher velocities and this evaluation needs to be included in the assessment of transmitral gradients. One major drawback of routine TEE in children is the need for deep sedation or general anesthesia. TEE is a reliable imaging modality to assist with the device closure of a PFO in the cardiac catheterization lab, particularly if a young patient is being anesthetized. In these cases, TEE is useful for determining the precise location (usually the cranial edge of the fossa ovalis), anatomy (slit-like, tunnel-like, aneurysmal or fenestrated) and the direction of shunting, and it can also assist in wire, sheath and device manipulations [29,30,31]. TEE is helpful to evaluate rarely for extension of vegetation in endocarditis, thrombus or left atrial myxoma. TEE is also a useful adjunct to procedures in which the PFO is traversed or dilated or an atrial shunt is created to allow access to the LA to obtain left-sided hemodynamic data and to perform left-sided interventions to the mitral valve, pulmonary vein stents, etc.

One critical scenario in which a PFO needs to be looked for is in any kind of dilated cardiomyopathy (Figure 3), in which atrial distension is associated with decreased myocardial perfusion. The presence of a PFO or the creation of an atrial level shunt can decrease atrial distension and encourage myocardial recovery in some cases, especially after the placement of an extracorporeal membrane oxygenator (ECMO) [32]. Left ventricular assist device (VAD) implantation is another option for some pediatric patients with refractory heart failure. In these patients, the implantation of a VAD abruptly decreases diastolic filling pressures and can cause right-to-left shunting with profound systemic hypoxemia. The pre-VAD TEE diagnosis of a PFO may be challenged by left atrial distension compressing the defect, which may become more apparent when the LA filling pressures decrease after the VAD placement. A PFO is closed at the time of VAD placement if its presence is detected in time; otherwise, device closure may be subsequently needed [33].

Advanced imaging techniques for the diagnosis of PFO: Other alternative modalities are rarely employed in the evaluation of PFOs in certain settings, particularly in pediatric patients with cryptogenic strokes. Transcranial Doppler (TCD) echocardiography allows the noninvasive diagnosis of a right-to-left shunt caused by a PFO by detecting bubble signals in the middle cerebral artery after the injection of agitated saline into the antecubital vein [34]. The authors of this study have reported excellent sensitivity and specificity of 97% and 93%, respectively, in detecting PFO in stroke patients. It is also recommended that that the injection be given at rest and accompanied by the Valsalva maneuver if necessary [35]. The principal limitations of TCD are that it only indicates the existence of right-to-left shunting, it does not distinguish between intracardiac and other extracardiac shunting and it provides no anatomical information about the PFO at all. Cardiac magnetic resonance (CMR) is used for the evaluation of complex CHD anatomies; many times, the PFO is visualized. In adults, CMR was found to be inferior to TEE in the detection of right-to-left shunting and in identifying atrial septal aneurysm, but recent CMR studies with ultrafast software technology have shown improved sensitivity compared to TEE for the diagnosis of PFO in adults [36,37]. Three-dimensional echocardiography (3D Echo) has been reported to accurately depict the anatomy of the PFO and atrial septum, but experience in children is limited [38]. Intracardiac echography (ICE) is rarely used during interventional cardiac catheterization in children and adolescents [39]. The advantage of ICE is that it may avoid the need for general anesthesia and is reported to be a reasonable alternative to TEE for the catheter-based procedure in children. This technique enables the adequate characterization of the shunting across the PFO and the better visualization of septal rims, especially in the case of the tunnel type PFO during percutaneous closure device deployment [40]. Limitations include the cost of the probe, the need for second venous access to use a large-sized catheter (at least 8 Fr) and the need for an experienced operator. This procedure can also be associated with rare complications, which include vascular injury, cardiac perforation and atrial arrhythmias.

## 5. PFO-Associated Medical Diseases

The left-to-right shunt across the PFO is trivial and not hemodynamically significant, and no treatment is necessary for otherwise asymptomatic children. However, PFOs in the presence of CHDs can be a source of the right-to-left shunt producing paradoxical embolism with resultant cerebrovascular accidents (CVAs)/transient ischemic attacks (TIAs), brain abscess or other problems such as migraine, Caisson’s disease or platypnea-orthodeoxia syndrome.

### 5.1. Residual PFOs in Previously Treated Complex CHD

Systemic desaturation and paradoxical embolism can occur because of the right-to-left shunt via PFO in patients who have been previously treated for complex CHD. It has been reported that the PFO should be closed when there are low oxygen saturations, preferably with a trans-catheter device [41]. As an example, in some TOF patients, when the PFO is left for the benefit of immediate post-operative management, this can sometimes result in paradoxical embolism due to elevated right side filling pressure because of the diastolic dysfunction of the RV, and the PFO will need to be closed [42].

### 5.2. Cerebrovascular Accidents (CVAs)/Transient Ischemic Attacks (TIAs)

The PFO ‘‘triad’’ has been reported, which combines raised RA pressure, a venous source of thrombosis and the presence of a PFO as the requirements to determine if paradoxical embolism is a potential cause of CVA or TIA [43]. The meta-analysis of pediatric studies revealed that there was no defining role of a PFO in pediatric stroke [44,45]. There is an interaction of risk factors for hypercoagulation, arteriopathy and genetic factors that contribute to most strokes in children with PFOs [46,47,48,49,50]. An atrial septal aneurysm in the adult patient population is defined as a risk factor for stroke, but not during childhood [51]. A PFO could be a risk factor for stroke in children with sickle cell anemia because they are predisposed to thrombosis and elevations of right heart pressure, which could promote paradoxical embolization [52]. Based on current guidelines from the European position paper, primary prevention measures are not recommended for cryptogenic stroke in patients with PFO [43]. Secondary prevention measures include medical treatment (antiplatelet drugs and anticoagulants) and percutaneous or surgical closure of the PFO after an event occurs in order to prevent the recurrence of TIAs or CVAs. To date, no randomized study has compared medical treatment with percutaneous device closure. There is a lack of data comparing antiplatelet drugs and anticoagulant treatments to make evidenced-based recommendations in children, unlike adults. Therefore, the choice of treatment for PFO should be made on an individual basis, assessing the risks and benefits for each patient.

### 5.3. Migraine

The mechanism by which PFO may promote migraine remains speculative. One postulated mechanism is transient hypoxemia, due to the right-to-left intracardiac shunting of blood through the PFO causing microinfarcts in the brain, leading to an increased tendency towards migraine. Another theory is that blood components which otherwise would be screened out within the lungs are shunted from the RA to the LA and thus bypass the pulmonary vascular bed and are directed towards the eyes and brain; they may serve as a vaso-constrictor, which generates migraines. These speculations and other clinical observations have sparked interest in the potential usefulness of PFO closure for migraine prevention. Several non-randomized studies have described the association of migraine with right-to-left shunting across the PFO and improved migraine symptoms after PFO closure in children [12,53,54,55,56]. The only prospective, randomized, double-blinded study, the Migraine Intervention with STARFlex Technology (MIST) trial in 2008, failed to reach the primary endpoint of the complete cessation of migraine symptoms for which the study was underpowered. However, the study found a much higher incidence of PFO with a right-to-left shunt in migraine with aura in patients than reported in the general population. The study did not support the efficacy seen in previous observational reports [53,54,55,56]. Recently, the Percutaneous Closure of PFO in Patients with Migraine (PREMIUM) trial did not meet the primary endpoint of reduction in the responder rate in patients with a frequent migraine after the closure of PFO [57]. Both the MIST and PREMIUM trials showed no relationship between migraine and shunting across the PFO.

### 5.4. Platypnea-Orthodeoxia Syndrome

This is a relatively uncommon but serious syndrome of systemic hypoxia and breathlessness in the upright position caused by the interatrial or intrapulmonary shunting of either a PFO or an ASD [58,59]. Symptoms of platypnea-orthodeoxia are thought to be due to positional right-to-left shunting. Standing upright can stretch the PFO, thus allowing more streaming of venous blood from the inferior venacava through the defect, whether or not a persistent Eustachian valve coexists. The diagnosis of a PFO with platypnea-orthodeoxia syndrome can be confirmed during a tilt-test, measuring arterial saturation, and a contrast echocardiography showing right-to-left shunting [59]. Platypnea-orhtodeoxia has also been reported as a complication in Fontan patients with pulmonary arteriovenous fistulas and/or the presence of a Fontan fenestration communicating with the RA [60]. The treatment for this syndrome is closure of the PFO or an intra-atrial shunt using a percutaneous device [11,61].

### 5.5. Role of a PFO in Exacerbating Hypoxemia 

PFO-mediated hypoxemia can occur when there is RA to LA shunting especially during exercise. Patients with a right-to-left shunt across the PFO may have profound hypoxemia out of proportion to underlying primary lung disease. In a subset of these patients who have normal right-sided pressures, percutaneous PFO closure may result in the marked improvement of their symptoms [62]. The presence of a PFO is about four times more frequent and hypoxemia is more severe in patients who are susceptible to high-altitude pulmonary edema than those living at high altitudes, who are resistant to this condition [63]. Interestingly, the presence of a PFO did not impact survival in primary pulmonary hypertension patients in one study [64]. The study’s authors speculated that the incidence of PFO increases in pulmonary hypertension patients with a more dilated and dysfunctional RV rather than it is congenital.

### 5.6. Decompression (Caisson’s) Syndrome

The association between PFO and decompression illness suffered by divers and high-altitude pilots who rapidly transition from high- to low-pressure environments has been documented [65]. The sudden decrease in pressure if the ascent from depth in the case of divers is too rapid results in the formation of nitrogen bubbles within tissues, and gas bubbles enter the arterial circulation, which can result in vessel occlusion. This can produce a range of symptoms, including muscle and joint pain, headache, dizziness, fatigue, rash, paresthesia, breathing difficulties, confusion, motor incoordination and paralysis [66,67]. A longitudinal, non-randomized follow-up study showed a reduction in both symptomatic neurological events and total brain lesions among recreational divers with PFO and decompression illness who had PFO closure, compared with those continuing to dive without closure [68]. According to this study, a PFO closure could be recommended for professional divers. However, there is controversy about whether young adults who are aviators, astronauts and scuba divers should undergo routine screening for PFO.

### 5.7. Systemic Embolism

Paradoxical systemic embolism through PFOs resulting in renal and myocardial infarctions have been reported [43]. However, there is no evidence from randomized controlled trials that closure of the PFO is protective against systemic embolization in any patients. The presence of a Eustachian valve (Figure 4A) and Chiari’s network (Figure 4B) may favor and facilitate paradoxical embolism [69,70]. However, unlike in adults, atrial septal aneurysm (Figure 4C) is not considered as a risk for thromboembolism in children [51].

## 6. Management of PFO

Currently, there are two approaches to closing a PFO: surgical and transcatheter closures. Surgical closure can be done simultaneously if other surgical interventions are necessary; otherwise, most cardiologists prefer the transcatheter closure of an isolated PFO in special circumstances such as platypnea-orhtodeoxia or when a PFO is causing the exacerbation of systemic hypoxia. Currently available percutaneous devices are the Amplatzer PFO occluder (St. Jude Medical, Inc., Golden Valley, MN, USA) and the Gore Cardioform Septal Occluder and Gore HELEX^®^ devices (W.L. Gore and Associates, Flagstaff, AZ, USA). Other therapeutic options are under development and include HeartStitch PFO (Sutura Inc, Fountain Valley, CA, USA), or BioTREK (NMT Medical, Boston, MA, USA) [71]. There is a lack of scientific evidence for the need to close PFOs or the use of antiplatelets or anticoagulants in asymptomatic children. The treatment of PFOs in patients with associated clinical syndromes is discussed under each specific condition. In children, there is no RCT to compare antiplatelets/anticoagulants with device therapy for the treatment of PFOs associated with CVA or TIA. Future trials should include children with migraine, decompression syndrome and other medical conditions associated with PFO in order to explore the benefits of PFO closure using percutaneous devices versus using medical therapy.

## 7. Conclusions

Patent foramen ovale in infants and children is a frequent finding but, in most patients, it presents no clinical implications. Due to the controversy regarding the clinical significance of PFOs in many diseases described in this review and the role of PFOs in the settings of dilated cardiomyopathy and pulmonary hypertension, and with the wide use of ECMO and VAD for pediatric decompensated heart failure, increased attention has been directed towards the presence of PFO in last few years. Generally speaking, an interdisciplinary and personalized approach is required for the management of PFO in certain special circumstances during fetal life and in infants and children, as outlined in this review.

## Figures and Tables

**Figure 1 medsci-08-00025-f001:**
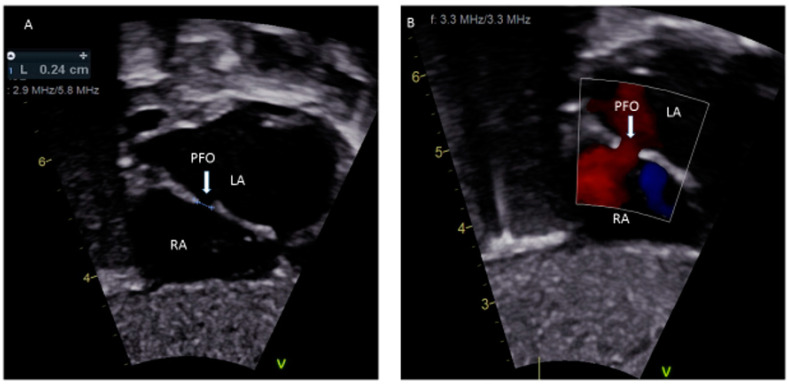
(**A**) Sub-costal view showing a 2.4 mm PFO; (**B**) Color Doppler demonstrating left to right shunting.

**Figure 2 medsci-08-00025-f002:**
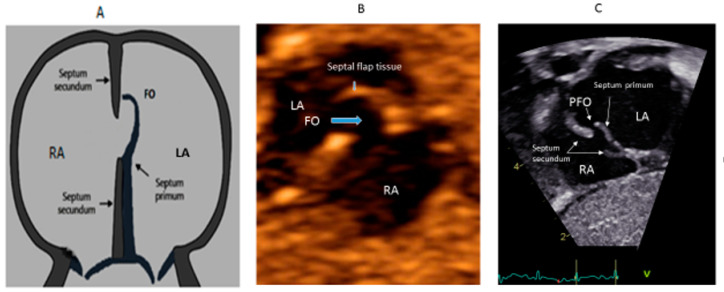
Correlation between development of atrial septum (**A**), Fetal (**B**) and post-natal anatomy (**C**).

**Figure 3 medsci-08-00025-f003:**
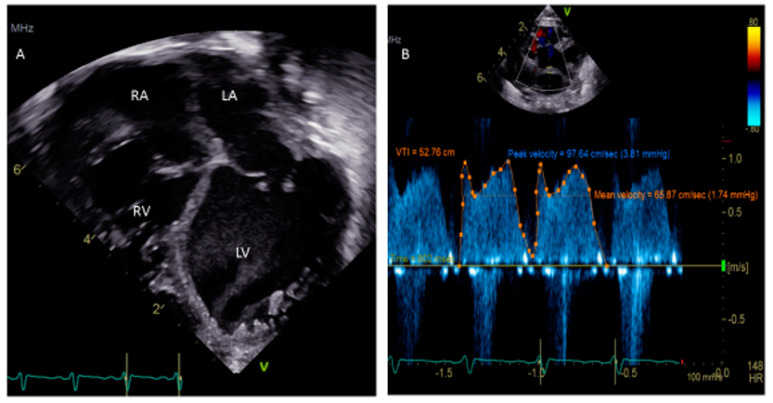
(**A**) Apical 4-chamber view showing dilated cardiomyopathy in a 5 week old child. (**B**) Doppler of PFO showing peak velocity 3.81 mm Hg and mean 1.74 mm Hg.

**Figure 4 medsci-08-00025-f004:**
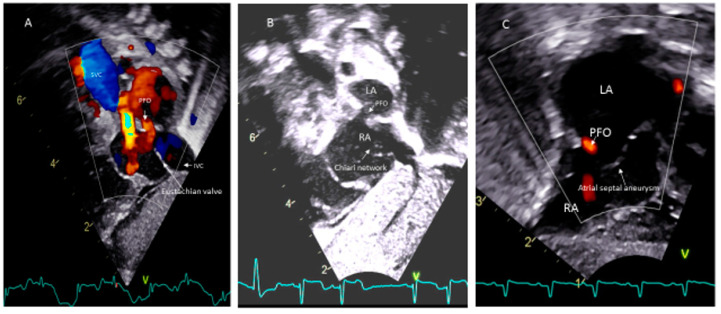
PFO associated with (**A**) Eustachian vale; (**B**) Chiari network; (**C**) Atrial septal aneurysm.

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
