# Peer review of "Patent Foramen Ovale in Fetal Life, Infancy and Childhood"

_medsci, 2020, doi:10.3390/medsci8030025_

Round 1

Reviewer 1 Report

In this manuscript, Bibhuti B Das reviewed the presence and diagnosis of patent foramen ovale at different ages. The author also discussed the potential association of PFO with medical diseases and current approaches for disease treatment. Overall, the first part of this this manuscript about PFO diagnosis at different ages is well-organized with proper and sufficient summarization. The last part of this article about PFO associated diseases and its management can be improved.

Some suggestions:

Section 5.1, TOF (Tetralogy of Fallot?) is not defined. More explanation about why PFO could cause paradoxical embolism in TOF patients can help complete this section.

The potential mechanisms that PFO induces diseases are not fully discussed in Section 5. For example, in section 5.3, though the mechanism by which PFO may promote migraine remains unclear, several hypotheses have been proposed like genetic correlation or by vasoactive substances which otherwise would be screened out within the lungs are permitted by the PFO reach the brain etc. Add some discussion about mechanisms could help understand the connection of PFO with disease.

Section6. The author discussed surgical and transcatheter closure treatment for PFO, but not the medical therapy. As in some cases, PFO may not necessarily be pathogenic, treatment like antiplatelet therapy could remain the treatment of choice for some associated diseases. I suggest adding some discussion about treatment selection with different conditions.  

Author Response

In this manuscript, Bibhuti B Das reviewed the presence and diagnosis of patent foramen ovale at different ages. The author also discussed the potential association of PFO with medical diseases and current approaches for disease treatment. Overall, the first part of this this manuscript about PFO diagnosis at different ages is well-organized with proper and sufficient summarization. The last part of this article about PFO associated diseases and its management can be improved.

Some suggestions:

Section 5.1, TOF (Tetralogy of Fallot?) is not defined. More explanation about why PFO could cause paradoxical embolism in TOF patients can help complete this section.

  • Thank you. TOF is defined.

The potential mechanisms that PFO induces diseases are not fully discussed in Section 5. For example, in section 5.3, though the mechanism by which PFO may promote migraine remains unclear, several hypotheses have been proposed like genetic correlation or by vasoactive substances which otherwise would be screened out within the lungs are permitted by the PFO reach the brain etc. Add some discussion about mechanisms could help understand the connection of PFO with disease.

  • Thank you for your useful comments. The potential mechanisms of PFO for each disease conditions are added.

Section6. The author discussed surgical and transcatheter closure treatment for PFO, but not the medical therapy. As in some cases, PFO may not necessarily be pathogenic, treatment like antiplatelet therapy could remain the treatment of choice for some associated diseases. I suggest adding some discussion about treatment selection with different conditions.  

  • Medical therapy such as antiplatelet are added in the treatment section.

Reviewer 2 Report

Even in a review article, method to survey literature should be given.

Author cited previous researches on risk factors of or intervention to PFO and their associated symptoms. But there is no description on relative risk, absolute risk and statistical significance. For better understanding among readers, the information is necessary.

Figures: Labels should be set in a unified format (font, size, etc.).

Author Response

Even in a review article, method to survey literature should be given.

  • The methods used to do a systematic search in PubMed and Cochrane database is added in the revised manuscript.

Author cited previous researches on risk factors of or intervention to PFO and their associated symptoms. But there is no description on relative risk, absolute risk and statistical significance. For better understanding among readers, the information is necessary.

  • The relative risk, absolute risk and statistical significance are available mostly for risk of cryptogenic stroke in adults. In children: the studies are small and not able to provide any relative risk or absolute risks or odds of stroke/TIA, migraine, or other rate associations. We have mentioned that it is difficult to give absolute benefits or risks in children due to lack of randomized control trials. It is not possible to extrapolate the adult studies as there is significant differences in physiology contributing to thrombus formation and paradoxical embolism.

Figures: Labels should be set in a unified format (font, size, etc.).

  • We used uniform Font and size in the labels of Figures.

Round 2

Reviewer 2 Report

I'm satisfied with the revision.

Only small correction needed is,

Line 41: correct 'RTC' to 'RCT'.